# Compound flood potential from storm surge and heavy precipitation in coastal China: dependence, drivers, and impacts

Jiayi Fang[1, 2], Thomas Wahl[3], Jian Fang[4], Xun Sun[1], Feng Kong[5], Min Liu[1]

[1]Key Laboratory of Geographic Information Science (Ministry of Education), School of Geographic Sciences, East China Normal University, Shanghai, 200241, China
[2]School of Environmental and Geographical Sciences, Shanghai Normal University, Shanghai, 200234, China
[3]Department of Civil, Environmental, and Construction Engineering and National Center for Integrated Coastal Research, University of Central Florida, 12800 Pegasus Drive, 32814 Orlando, USA
[4]College of Urban and Environmental Science, Central China Normal University, Wuhan, 430079, China
[5]College of Humanities and Development Studies, China Agriculture University, Beijing, 100083, China

*Correspondence to*: Jiayi Fang (jyfang822@foxmail.com; jyfang@geo.ecnu.edu.cn)

**Abstract.** The interaction between storm surge and concurrent precipitation is poorly understood in many coastal regions. This paper investigates the potential compound effects from these two flooding drivers along the coast of China for the first time by using the most comprehensive records of storm surge and precipitation. Statistically significant dependence between flooding drivers exists at the majority of locations that are analysed, but the strength of the correlation varies spatially and temporally and depending on how extreme events are defined. In general, we find higher dependence at the south-eastern tide gauges (TGs) (latitude < 30°N) compared to the northern TGs. Seasonal variations in the dependence are also evident. Overall there are more sites with significant dependence in the tropical cyclone (TC) season, especially in the summer. Accounting for past sea level rise further increases the dependence between flooding drivers and future sea level rise will hence likely lead to an increase in the frequency of compound events. We also find notable differences in the meteorological patterns associated with events where both drivers are extreme versus events where only one driver is extreme. Events with both extreme drivers at south-eastern TG sites are caused by low-pressure systems with similar characteristics across locations, including high precipitable water content (PWC) and strong winds that generate high storm surge. Based on historical disaster damages records of Hong Kong, events with both extreme drivers account for the vast majority of damages and casualties, compared to univariate flooding events, where only one flooding driver occurred. Given the large coastal population and low capacity of drainage systems in many Chinese urban coastal areas, these findings highlight the necessity to incorporate

compound flooding and its potential changes in a warming climate into risk assessments, urban planning, and the design of coastal infrastructure and flood defences.

**Keywords**. Compound flood, Storm surge, Precipitation, China

## 1 Introduction

Floods are among the costliest and deadliest disasters globally (Hu et al., 2018). In recent years, a series of devastating compound flooding events occurred, such as Hurricane Isaac in 2012, Typhoon Haiyan in 2013, Hurricanes Irma and Florence in 2018, and Typhoon Lekima in 2019. Despite improvements in flood defences, flood forecasting, and warnings, these flood events caused devastating impacts, in parts due to the limited understanding of compound floods in coastal regions. Flooding along the coast can arise from three main sources: 1) extreme sea levels (comprised of storm surge, high astronomical tides, and/or waves (coastal flood)); 2) river discharge (fluvial flood); and 3) direct surface run-off from rainfall (pluvial flood) (Hendry et al., 2019). Floods in coastal areas are frequently caused by more than one driver and the impacts when they coincide are often much greater than from either flood driver occurring in isolation (Leonard et al., 2014; Zscheischler et al., 2018; Hao et al., 2018). Exploring the probabilities of compound flood events and understanding their driving processes is important for flood mitigation and risk reduction in a warming climate (Wahl et al., 2015).

A growing number of studies investigated compound flooding in recent years. At the global scale, dependence between storm surge and river discharge has been investigated based on observational data (Ward et al., 2018) and model hindcasts (Bevacqua et al., 2020; Couasnon et al., 2020). The relationship between storm surge and wind waves was assessed by Marcos et al. (2019). At the regional scale, compound flood assessments have been undertaken for Australia (Zheng et al.,2013; Zheng et al.,2014; Wu et al., 2018), the USA (Wahl et al., 2015), the UK (Svensson and Jones, 2002, 2004; Hendry et al., 2019), and Europe (Petroliagkis et al., 2016; Paprotny et al., 2018; Bevacqua et al., 2019; Ganguli and Merz, 2019). Other studies focused on specific locations, such as the Netherlands (van den Hurk et al., 2015); Fuzhou, China (Lian et al., 2013); Taiwan, China (Chen and Liu, 2014), or the North Sea (Khanal et al., 2019). Most of these studies investigated the dependence between two hazards, such as storm surge

and river discharge, storm surge and waves, or storm surge and rainfall, thus assessing compound flood potential as opposed to actual compound flood risk.

For China, a comprehensive regional assessment of the compound flooding potential is currently missing. Low-lying coastal areas (elevation less than 10 m) in China only account for 2% of the national land, but account for more than 12% of the national population (Liu et al., 2015; Fang et al., 2020). At the same time, these areas are experiencing frequent coastal disasters from TCs and storm surges, among others. Coastal flooding has caused more than US\$ 71 billion direct economic losses and 4,376 fatalities in China from 1989 to 2014 (Fang et al., 2017). Flood risk is likely increasing in China due to climate change (most notably sea level rise), as well as human factors (e.g. human-induced subsidence) (Fang et al., 2020; Jiang et al., 2020; Fang et al., 2021; Wu et al., 2005; Wu et al., 2017). Meanwhile, fast urbanisation in China has led to more people and economic assets exposed to hazards (Fang et al., 2018; Du et al., 2018), and has also prompted irrational urban planning, increased areas of urban impervious surface, and low capacity drainage systems (Cheng, 2020). For example, the capacity of the local drainage system of Shenzhen City is designed to drain the surface runoff associated with a 2-year return period (or 50% annual exceedance probability) (Urban Planning & Design Institute of Shenzhen, China, 2008). As drainage facilities are often under-designed and/or have not been upgraded, surface runoff during storms frequently exceeds the drainage capacity resulting in flooding damages in low-lying areas (Qin et al., 2013). Despite the relevance of compound flooding for coastal China, the associated probabilities and driving mechanisms have not been explored at broad spatial scales at the national level.

A limited number of studies have assessed different aspects of compound flooding for China. Lian et al. (2013) and Xu et al. (2014) investigated the joint probability, using copulas, of extreme precipitation and storm tide and associated changes for Fuzhou city. Both studies showed that the joint impacts from surge and precipitation were much higher than from each individually; this is currently ignored in the design of flood defences. Xing et al. (2015) analysed joint return periods of precipitation and runoff in the upper Huai River Basin in China. Ye and Fang (2018) estimated compound hazard severity of TCs considering extreme wind and precipitation. Changes in storm surges and precipitation in China have also been investigated separately, showing significant increases in extreme precipitation in parts of the southwest

and south China coastal areas (Zhai et al., 2005). Similarly, significant increases in sea level extremes have been reported (Feng et al., 2014; Feng et al., 2019), and attributed to both changes in mean sea level (MSL) and in the wind driven storm surge component (Feng and Tsimplis, 2015). However, these previous studies were mostly local, they neglected seasonal characteristics, and weather circulation patterns driving compound events were not assessed. In this study, we use the most comprehensive records of storm surge and precipitation to investigate dependences between the two flooding drivers and incidences of joint occurrences along with the synoptic weather patterns causing events where compound flood potential is high versus events where it is low. We also conduct a preliminary analysis of the potential impacts in terms of recorded losses and fatalities of past compound events caused by TCs.

In this context our four main objectives are: 1) assess the dependence between storm surge and precipitation for different thresholds for the full year and the summer and TC seasons; 2) examine the role of sea level rise in escalating compound flood potential; 3) identify large scale weather systems leading to events with high versus low compound flood potential; and 4) explore possible contribution of compound flooding during past TC events.

## 2 Data

Most tide gauge (TG) data are kept confidential in China; thus, we obtained hourly sea level data of 11 TGs with at least 20-year lengths along the Chinese coast from the University of Hawaii Sea Level Center (Caldwell et al., 2015). Locations of TGs and the time series' lengths are shown in Fig. 1. The stations are located south of the Shandong peninsula in China, where TC impacts are most severe (He et al., 2015). Nine of the 11 TG stations have about 20 years of data (1975-1997), Xiamen and Hong Kong have 46 years (1954-1997) and 52 years (1962-2014), respectively.

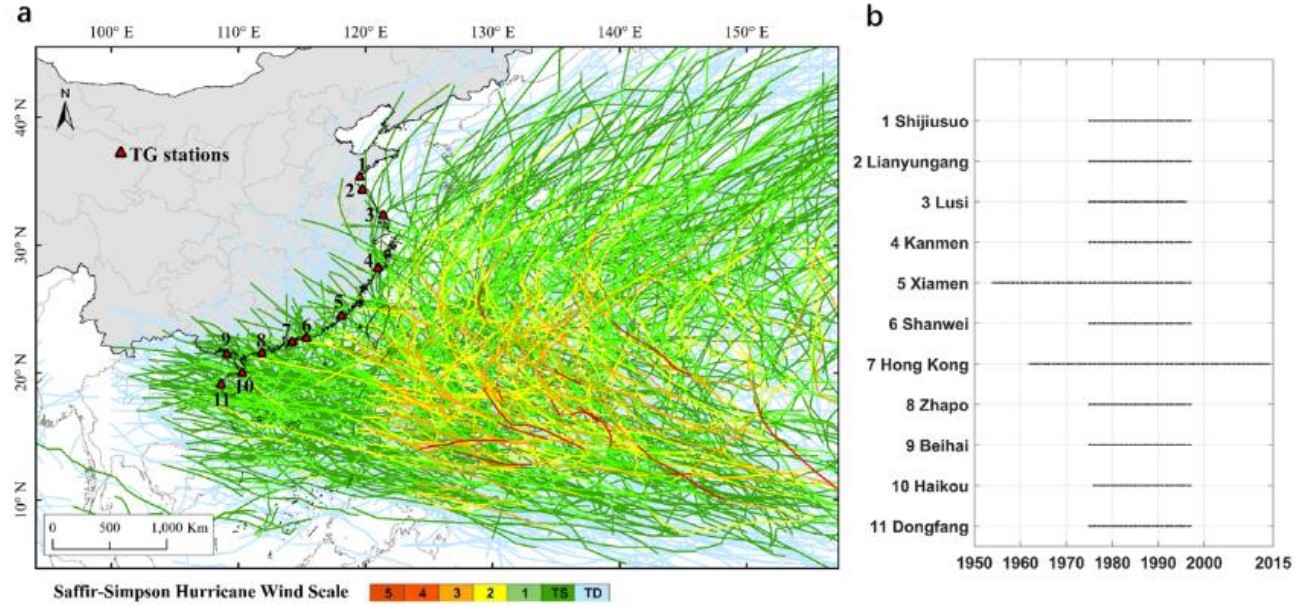

Fig.1 (a) Locations of 11 tide gauges and historical TC tracks for different intensities (only 1975-1997 shown here); (b) time periods covered by hourly sea level data at the 11 tide gauges.

Storm surge is extracted using the MATLAB t_tide package (Pawlowicz et al., 2002) by applying a year-by-year harmonic tidal analysis with 67 constituents. This also effectively removes the MSL influence
including the long-term trend in MSL as well as the year-to-year and decadal variability (Wahl et al., 2015). The data has been checked for common errors and 75% completeness of each year is required. An offset of 1.02 cm in the Hong Kong data in 1986/1987 (due to changes in the TG location) is adjusted following Ding et al. (2002).

Daily cumulative precipitation records from 1951-2015 are collected from China Meteorological
Administration. The closest meteorological station is chosen to match each TG station, and the distance between them is less than 25 km for 9 out of 11 TGs (TG2 with 29 km and TG8 with 34 km). The time series of precipitation observations are usually longer and more complete than TG observations; thus TG data availability determines the lengths of overlapping periods available for the dependence analysis presented here.

To identify weather patterns typically associated with events that have high compound flood potential versus those that have low potential, sea level pressure (SLP), precipitable water content (PWC), and wind fields are used from the Twentieth Century Reanalysis Project Version 2c (Compo et al.,2011).

To assess the impacts of past TC events where both flooding drivers were extreme versus events where only one was extreme, we employ a damage database developed by Yap et al. (2015). It includes historical
TC records from 1951 to 2012. The database contains information of 853 TCs with direct normalized economic loss (in US$), death toll, and number of people affected.

**3 Methodology**

**3.1 Defining compound events and dependence analysis**

Compound events have been defined in different ways in the past, either based on impact information
(which we don't have) or based on the severity of the flooding drivers involved, assuming that at least one is extreme. In our study, we adopt the latter approach to account for the fact that the combination of storm surge and precipitation can exacerbate flood impacts in different ways (Wahl et al., 2015). First, both heavy precipitation and extreme sea levels (storm surge with high tides) can coincide, leading to more severe floods. This often happens during TC events. Second, impacts of a storm surge already
causing flooding will increase when significant precipitation occurs at the same time, although the precipitation itself may not be considered extreme. Third, a moderate storm surge can block freshwater water drainage and high precipitation occurring at the same time can lead to more severe flooding (as compared to the same rain event coinciding with low sea level). To illustrate the definition, time series of daily maximum storm surge are plotted against records of daily cumulative precipitation for Kanmen
(TG4) in Fig. 2. In the following analysis, three distinct cases are considered. First, Zone 1 shows the events with joint occurrence of high storm surge and heavy precipitation. Second, "Zone 1 + Zone 3" refers to high storm surge with or without heavy precipitation. Third, "Zone 1 + Zone 2" refers to heavy precipitation with or without high storm surge. These three combinations capture all of the above-mentioned mechanisms. We select extreme storm surge and the corresponding precipitation within ± 1

145 day of the surge, or select extreme precipitation and the corresponding storm surges within ± 1 day of the precipitation.

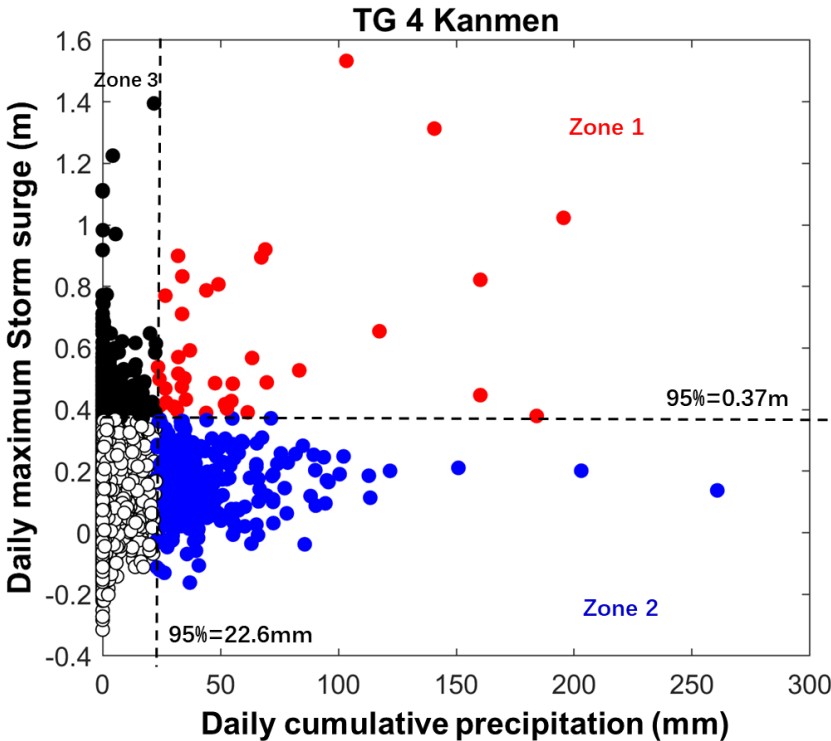

Fig. 2 Daily maximum storm surge plotted against daily cumulative precipitation for threshold of 95%. Red dots (plotted in Zone 1) show joint occurrence of high storm surge and heavy precipitation, whereas blue (Zone 2) and black (Zone 3) dots

define events where only one flooding driver is extreme.

We use the peaks over threshold (POT) method to select extreme events. The POT method refers to selecting events over a high threshold within a certain time span. The annual maximum approach is widely used for sampling extreme events. However, it would lead to small sample sizes here as time series of 9 out of 11 TGs in China only have around 23 years of data. Furthermore, the second or third largest values

in a given year may be larger than the annual maximum in another year (Coles et al., 2001; Arns et al., 2013). To test the sensitivity of the results to the threshold selection, we employ thresholds related to

eight percentiles ranging from 95% to 99.5%, i.e., 95%, 96%, 97%, 98%, 98.5%, 99%, 99.25% and 99.5%. Independence of the threshold exceedances is achieved using a declustering time of 3 days.

Kendall's rank correlation coefficient $\tau$ is employed to measure dependence between storm surge and precipitation. In "Zone 1 + Zone 3", storm surges sometimes could occur without any precipitation, and this leads to ties (i.e., several zero values) affecting the dependence analysis. We use the same method as suggested in Kojadinovic and Yan (2010) and Wahl et al. (2015) by assigning ranks randomly, repeating the procedure 100 times and calculating the average rank correlation. To better understand the influence of seasonality, dependence is assessed for the full year as well as for summer (June to August) and the TC season (July to October).

### 3.2 Effects from sea level rise on compound flood potential

To test how the inclusion of sea level rise affects joint occurrences of flooding drivers, we count the occurrences between storm surge and precipitation with or without MSL. The effects of MSL are initially removed during the harmonic tidal analysis. We repeat the same sampling approach as outlined above but keep the MSL influence and extract surge events by only removing the tidal influence, i.e., total water level minus tide. Because of the inclusion of MSL and the resulting nonstationarity in the data, we can not carry out the dependence analysis in the same way as before. Thus, we simply count the number of pairs in Zone 1, "Zone 1 + Zone 3" and "Zone 1 + Zone 2" with and without MSL and compare them.

### 3.3 Weather patterns driving events with high and low compound flood potential

To investigate the meteorological patterns that drive events with high and low compound flood potential, we select a threshold of 98%. We consider events with high compound flood potential to fall into Zone 1 in Fig. 2. Events with low compound flood potential fall in Zone 2 or Zone 3 in Fig. 2. SLP, PWC, and wind fields from the days when the events occurred are selected, and averaged into composites to represent reference synoptic-scale weather patterns favouring compound flooding.

**3.4 Losses of past TC events where compound flood potential was high or low**

To quantify the differences in impacts caused by past TCs when compound flood potential was high or low, we use a TC damage database developed by Yap et al (2015) for Hong Kong as a case study. We identify events with high/low compound flood potential in the same way as we did for the synoptic weather type analysis, then match the days when the events occurred with records in the database; the 185 latter includes information on death toll, people affected, and economic losses.

**4 Results**

**4.1 Dependence between storm surge and precipitation and seasonal variation**

Fig. 3 shows the Kendall dependence between all pairs of daily maximum storm surge and daily cumulative precipitation (Fig. 3a), as well as pairs in Zone 1 (Fig. 3b), pairs in "Zone 1 + Zone 3" (Fig. 190 3c), and pairs in "Zone 1 + Zone 2" (Fig. 3d), respectively. Figs. 3b-d show the maximum dependence for thresholds running from 95% to 99.5%. As expected, the three cases where events above high thresholds are collated have relatively higher dependence compared to when all data are used. For Zone 1, most TGs show insignificant dependence indicating a limited number of joint occurrences. For "Zone 1 + Zone 3", south-east coastal China, which is more affected by TCs (Fig. 1), exhibits higher dependence 195 than the northern part. Overall, "Zone 1 + Zone 3" dependence is also higher than "Zone 1 + Zone 2" dependence and we identify more locations with significant dependence, 9 TGs in "Zone 1 + Zone 3" and 7 TGs in "Zone 1 + Zone 2", respectively.

Kanmen (TG4) and Haikou (TG10) show relatively high dependence for all cases. Shanwei (TG6) and Zhapo (TG8) show high positive dependence in Zone 1 and "Zone 1 + Zone 3", but insignificant 200 dependence in "Zone 1 + Zone 2", indicating that high storm surge is often accompanied by high rainfall but not the other way round. The opposite is true for Lusi (TG3), which has positive dependence in "Zone 1 + Zone 2", but insignificant dependence in "Zone 1 + Zone 3".

We also test impact of the thresholds (95% to 99.5%) which can influence the correlation (Supplementary Fig. S1). At most locations the dependence increases when higher thresholds are used to sample extremes.

There are exceptions however, for example, Haikou (TG10) in "Zone 1 + Zone 2" shows higher
dependence with a threshold of 99% than 99.5%. At some TGs dependence becomes insignificant due to
small sample sizes when thresholds are very high, indicating the trade-off between bias and variance in
the threshold selection. Thresholds for compound events are very localized and highly dependent on the
underlying data and various methods exist to select bivariate extremes (Salvadori, et al., 2016), we did
not compare those methods here as it would go beyond the scope of our study.

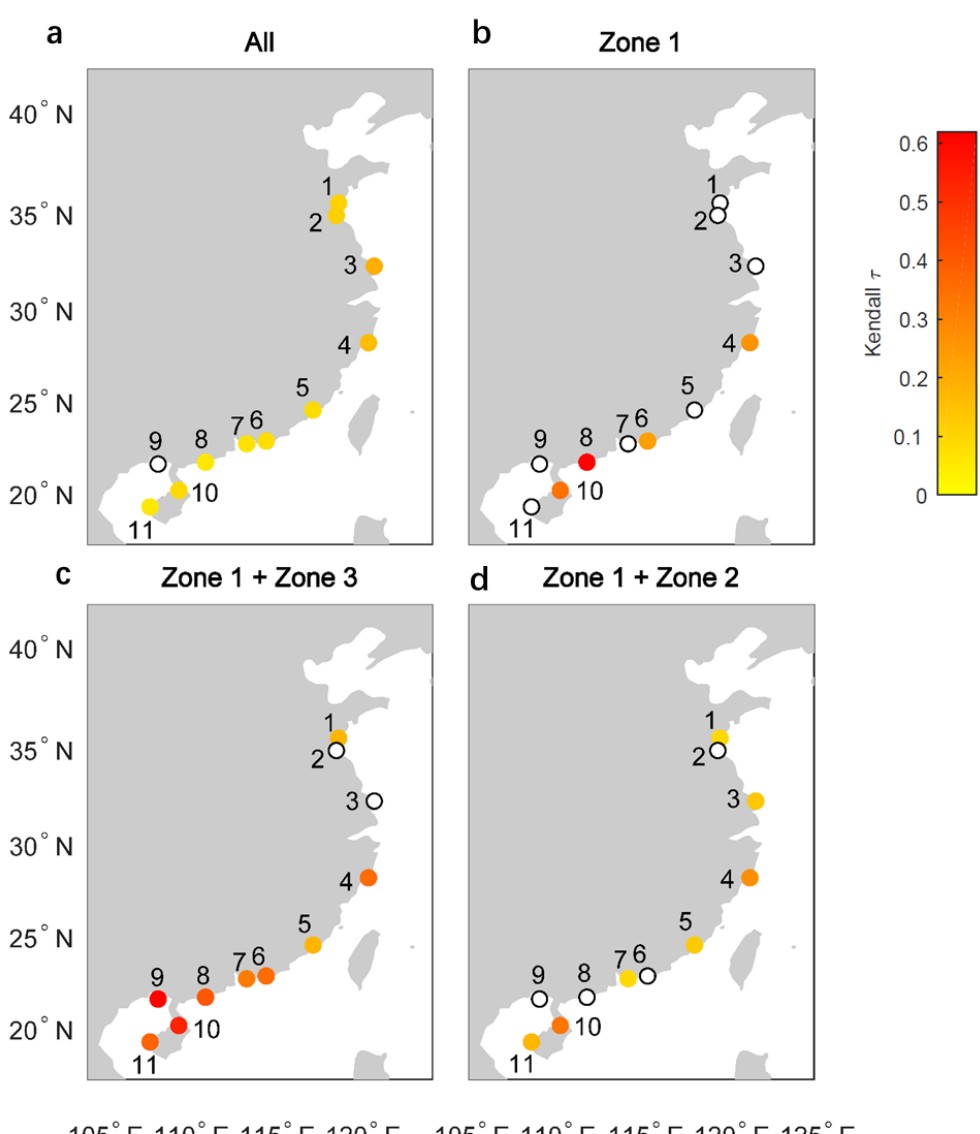

Fig. 3 Kendall dependence between storm surge and precipitation. a) daily maximum storm surge and daily cumulative precipitation; b) pairs in Zone 1; c) pairs in "Zone 1 + Zone 3"; d) pairs in "Zone 1 + Zone 2". b-d) shows the maximum dependence for thresholds running from 95% to 99.5%. White dots refer to insignificant dependence (10% level).

To better understand the timing of events with relatively higher compound flood potential, the influence of seasons is investigated. TCs are active over the western North Pacific during July to October (He et al., 2015). Thus, three periods are considered: TC season (July-October), summer (June-August) and whole year. The seasonal dependences are displayed in Fig. 4. It shows that dependences in "Zone 1 + Zone 3" and "Zone 1 + Zone 2" are stronger than Zone 1 dependence for all seasons. "Zone 1 + Zone 3" also
shows stronger significant dependence than "Zone 1 + Zone 2".

For "Zone 1 + Zone 3", multiple TGs show stronger dependence in summer and in the TC season compared to the whole year, such as Kanmen (TG4), Shanwei (TG6), and Hong Kong (TG7). Zhapo (TG8) shows insignificant dependence in "Zone 1 + Zone 2", while significant positive dependence is found in "Zone 1 + Zone 3". TGs from Kanmen (TG4) to Dongfang (TG11) with latitudes less than 30°N
are most affected by TCs, and show high dependence for "Zone 1 + Zone 3", especially in summer and in the TC season. Xiamen (TG5) is an exception, likely because Taiwan Island weakens the intensity of TCs before reaching Xiamen.

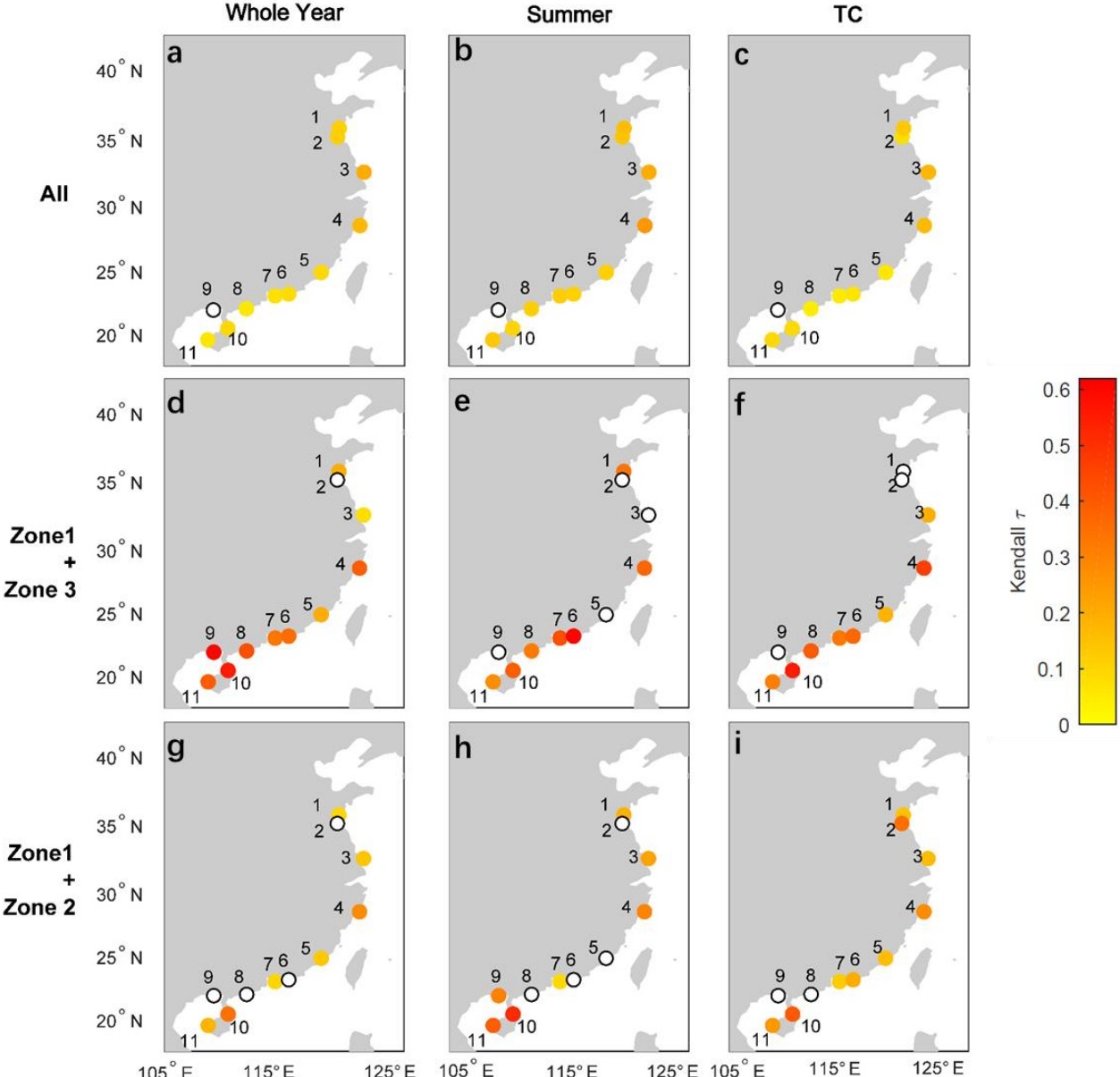

Fig. 4 Kendall dependence between storm surge and precipitation in summer, the TC season, and the whole year. a-c) daily maximum storm surge and daily cumulative precipitation, d-f) pairs in "Zone 1 + Zone 3", and g-i) pairs in "Zone 1 + Zone 2"; maximum dependence for thresholds running from 95% to 99.5% is shown. White dots refer to insignificant dependence (10% level).

Dependence also varies with the threshold selection when performing the seasonal analysis. To illustrate this, five TGs are selected as examples in Fig. 5. Dependence continuously increases with higher thresholds. Again, for some TGs, dependence becomes insignificant for high thresholds, especially when records are short. For some TGs, the dependences in TC season are similar with the whole year, like Shanwei (TG6) and Zhapo (TG8) in "Zone 1 + Zone 3", indicating that most events with compound flood potential occur in the TC season. For example, 62% compound events ("Zone 1 + Zone 3" in Fig. 2) for Shanwei (TG6) and 64% for Zhapo (TG8) occurred in the TC season.

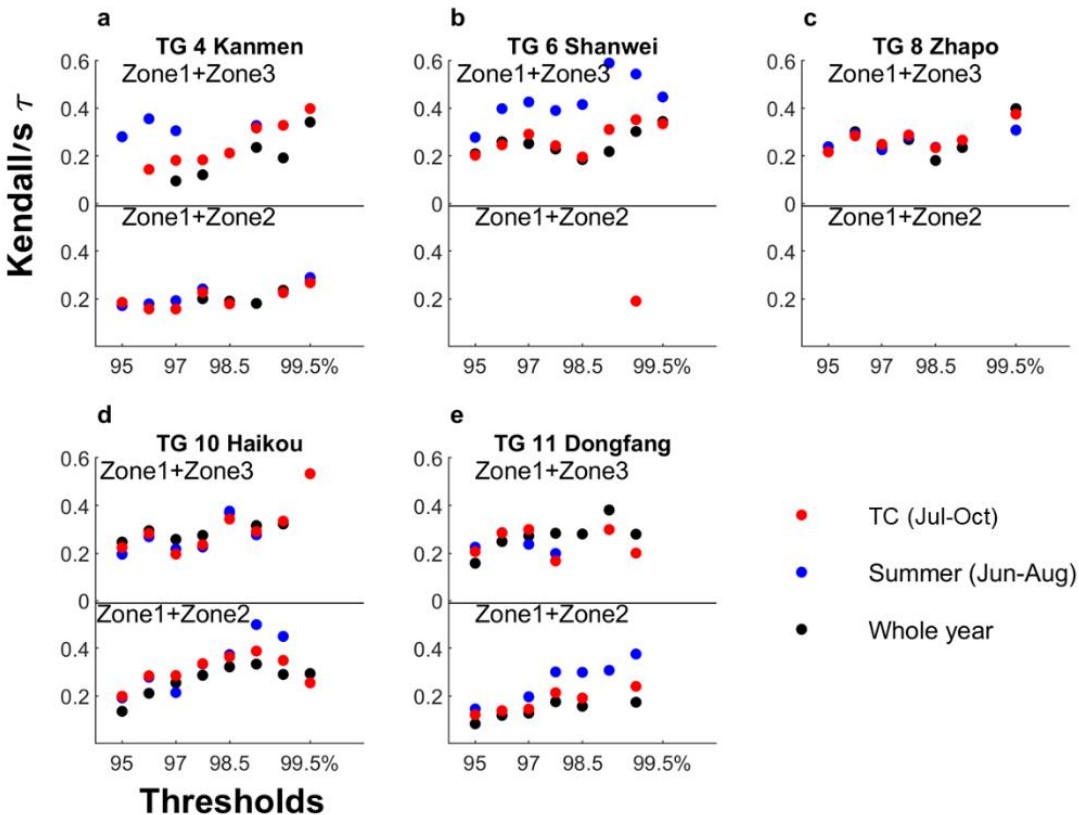

Fig. 5 Kendall dependence between storm surge and precipitation in summer, the TC season, and the whole year for thresholds running from 95% to 99.5%. a) Kanmen (TG4); b) Shanwei (TG6); c) Zhapo (TG8); d) Haikou (TG10); e) Dongfang (TG11).

South-east coastal China is not only affected by TCs, but also by summer monsoon precipitation from the Northwest Pacific Subtropical High. The summer monsoon brings continuous precipitation since June to

August in southern China. With the conincidence of frequent storm surge in July and August, it may explain the higher dependence in the summer compared to the TC season. It has been reported that an abrupt increase of intense TCs occurred in September after the mid-2000s for south China (He et al., 2016), which could affect the seasonality of compound events. However, from the results shown in this study, this pattern is not captured due to limited observation (most observations end in 1997).

## 4.2 Effects of sea level rise on compound flood potential

To better understand the effect of sea level rise on compound flood potential, we count occurrences between storm surge and precipitation with or without MSL; the differences are displayed in Fig. 6 for three cases. For Zone 1, 8 out of 11 TGs show an increase of joint occurrences when MSL influence is included, whereas the number at Haikou (TG10) remains unchanged. At Lianyungang (TG2) and Beihai (TG9), only one joint occurrence event is identified when MSL is removed, while this number increases to 22 and 26, respectively, with MSL included. Similar patterns are observed for "Zone 1 + Zone 3" and "Zone 1 + Zone 2" with most TGs showing increases of joint occurrences. For "Zone 1 + Zone 3", 6 out of 11 TGs show a relatively strong increase and remainder show a slight decrease. For "Zone 1 + Zone 2", more joint occurrences are observed at 10 out of 11 TGs. Lianyungang (TG2), Xiamen (TG5), Hong Kong (TG7) and Beihai (TG9) show the largest increases when MSL is included.

The results indicate that coastal China will experience an increasing frequency of events with high compound flood potential under future MSL rise. This is in line with Moftakhari et al. (2017) and Bevacqua et al. (2019), who also report that MSL rise will lead to more compound events. Sea level rise not only increases the probability of coastal flooding from storm surges (Buchanan et al., 2017), but also poses an additional threat for coastal communities susceptible to compound flooding. Meanwhile, other flood drivers, such as precipitation, river discharge and waves, can also exhibit nonstationarity leading to increased (compound) flood risk (Kundzewicz et al., 2019). Observations from the last five decades and numerical model studies (Lai et al., 2020) indicate a slowdown of TCs, which would likely favour more extreme rainfall during the events as compared to fast-moving TCs.

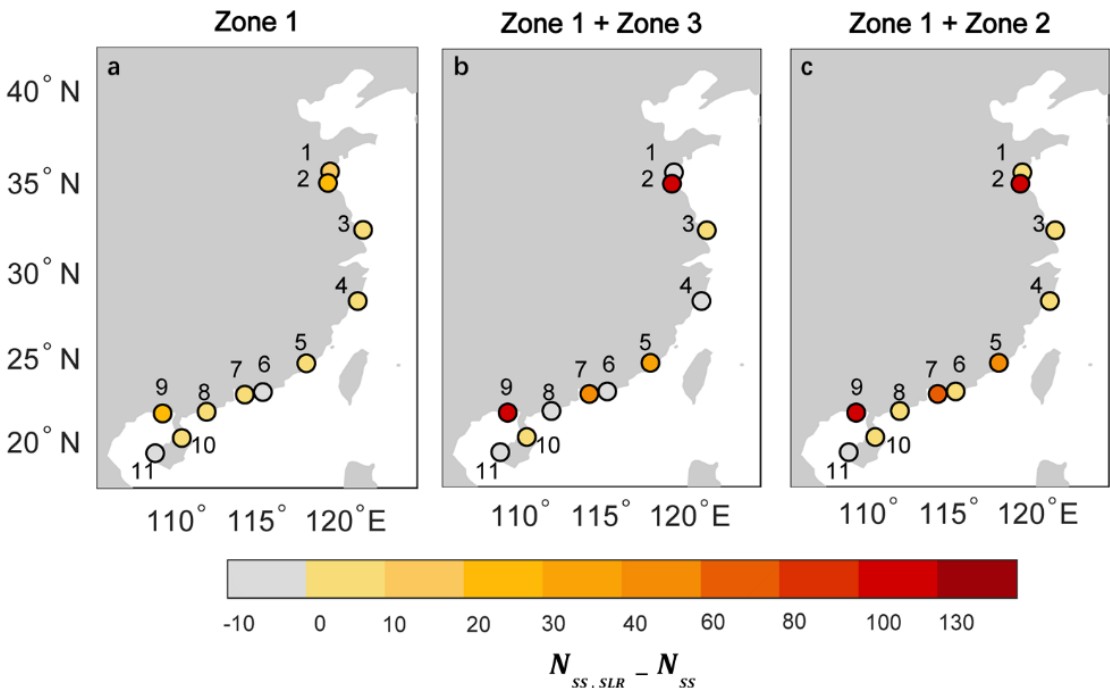

Fig. 6 Count of joint occurrences between storm surge and precipitation with/without mean sea level influence at threshold of 98%. $N_{SS+SLR}$ indicates joint occurrences considering historical sea level rise trend. $N_{SS}$ indicates joint occurrences after removing the effects of sea level rise. a) Zone 1; b) Zone 1 + Zone 3; b) Zone 1 + Zone 2.

### 4.3 Weather patterns driving events with high and low compound flood potential

We derived composite plots of synoptic conditions of SLP, PWC, and wind fields that drive joint occurrence events (both high storm surge and heavy precipitation; Zone 1 in Fig. 2) and events where only one driver was extreme (high storm surge or heavy precipitation, Zones 2 and 3 in Fig. 2) across coastal China. To illustrate the results, we focus on Kanmen (TG4) and Shanwei (TG6) on the east and south coast of China, which both have been frequently affected by TCs (Fig. 7 and Fig. 8). Results for the

other nine stations are shown in Supplementary Figs. S2- S10. Based on the 98% threshold we selected to identify joint occurrences, we identify 15 events for Kanmen (TG4) and 21 events for Shanwei (TG6), respectively.

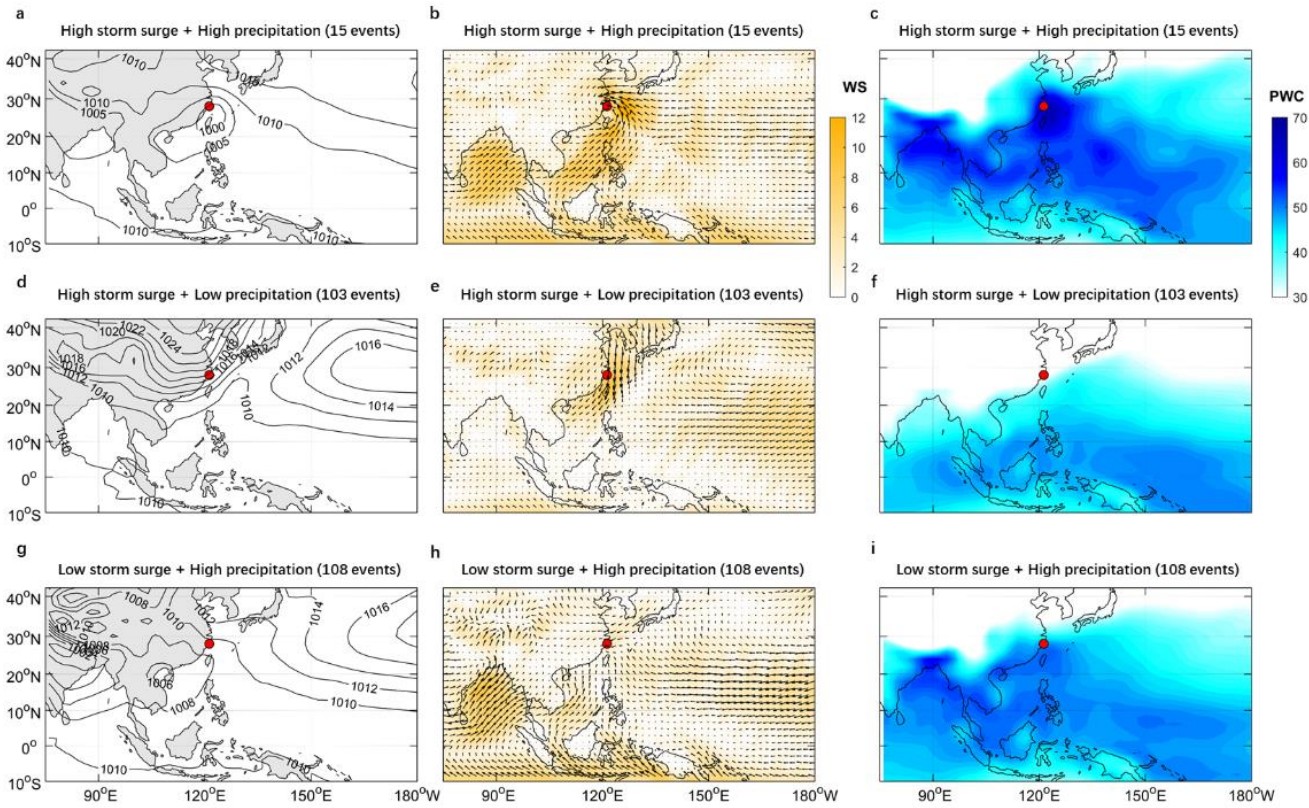

Fig. 7 Meteorology conditions for Kanmen (TG 4): (a, d, g) sea-level pressure (mbar), (b, e, h) wind speed (m/s) and direction (grey arrows), and (c, f, i) precipitable water content (PWC, kg m$^{-2}$) during (a, b, c) joint occurrence events with high storm surge and high precipitation (Zone 1), (d, e, f) for events with high storm surge and low precipitation (Zone 3), and (g, h, i) events with low storm surge and high precipitation (Zone 2).

The meteorological patterns in SLP, PWC, and wind fields are distinctly different across the three event types. At Kanmen (TG4), joint occurrence events are associated with a well-defined low-pressure system with strong east-west and south-westerly winds transporting moist air toward the south-eastern coast of China (Fig. 7a-c). Events with high storm surge and low precipitation exhibit a distinct pressure gradient along the coast (Fig. 7d). As expected, the wind speed is much stronger along the coast for this case (Fig. 7e) compared to the one where only precipitation is high (Fig. 7h), and the northern high wind drives moist air away from the site of interest. The differences in PWC patterns for with high and low compound

flood potential are more pronounced (Fig. 7c, f, i). Low storm surge and high precipitation events could also be caused by strong local convective rainfalls, which may not be captured here. There is low PWC for the type of only high storm surge events (Fig. 7f), while high PWC from the Bay of Bengal and cross-equatorial flow is observed for the other two types of events.

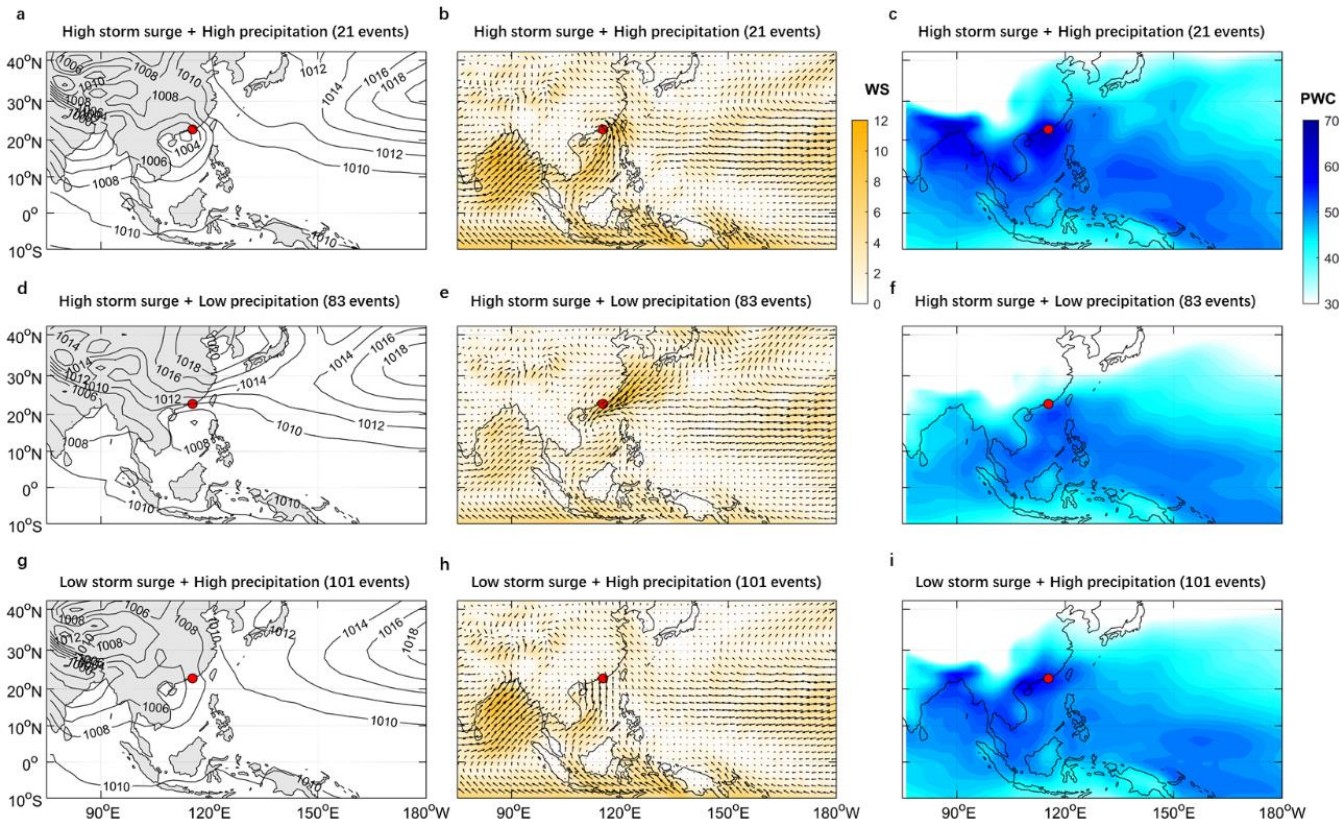

Fig. 8 Meteorology conditions for Shanwei (TG 6): (a, d, g) sea-level pressure (mbar), (b, e, h) wind speed (m/s) and direction (grey arrows), and (c, f, i) precipitable water content (kg m-2) during (a, b, c) joint occurrence events with high storm surge and high precipitation (Zone 1), (d, e, f) for events with high storm surge and low precipitation (Zone 3), and (g, h, i) events with low storm surge and high precipitation (Zone 2).

At Shanwei (TG 6), similarly to Kanmen, the meteorological patterns in SLP show a cyclone-structure

for both joint occurrence events and events with only high storm surge (Fig. 8a and 8d). For events with

high storm surge only, there is a distinct pressure gradient and strong wind speed (Fig. 8e). The PWC is low for the high storm surge events and high for joint occurrence and high precipitation only events (Fig. 8c, f, i). For precipitation only events, flows from the Bay of Bengal and cross-equatorial flow is observed, and south-eastern wind drives moist air to the site of interest (Fig. 8b and 7h).

The results for other stations are similar (Supplementary Figs. S2-S10). For joint occurrence events, synoptic weather patterns for south-eastern TG sites (latitude < 30°N) show similar low-pressure systems carrying intense PWC and causing strong wind. For northern TGs, such as TGs 1-3 (Supplementary Figs. S2- S4), the low-pressure systems are less developed compared to other TG sites. As most TCs make landfall along the south-eastern China coasts, their intensity decreases when they move from south to 315 north (see also Fig. 1).

### 4.4 Losses of past TC events where compound flood potential was high or low

Based on the flooding driver information (storm surge and precipitation), we identify 315 events in total for Hong Kong, including 44 events in Zone 1, 116 events for Zone 2 and 155 events for Zone 3 in Fig. 9. The damage database does not include information on all events that we identified, as it is a historical 320 TC disaster damage dataset, meaning that events that are not associated with a TC are excluded. None of the 168 events which are identified based on storm surge and precipitation data but not included in the damage database were joint occurrences where both flooding drivers were extreme. 68% of those unmatched events (115 events) are identified as low storm surge and high precipitation, and hence more likely related to convective rainfall events. It also indicates that not all those events led to significant 325 damages or the damages were not recorded. As shown in Fig. 9b, joint occurrences caused 227 deaths (average 5 deaths per event), affected 29,550 people (672 affected people per event), and led to US$ 221 million (average US$ 5 million per event) damages. Events with lower compound flood potential where only one driver was extreme caused 65 deaths (average 0.24 deaths per event), affected 6469 people (23.87 affected per event), and caused US$ 0.92 million (US$ 0.003 million) recorded damages. Hence, 330 joint occurrence events contributed 78% of the reported causalities, 82% of the people affected, and the vast majority of recorded damages. It is difficult to exactly quantify the contributions of compound flooding events to the impacts, as reported historical damage records could contain inaccuracies and

inconsistencies, such as various reported numbers from different sources and incomplete information. Furthermore, the TC damages not only result from flooding due to heavy rainfall and storm surge, but

also include damages from other effects, such as gale (strong wind). From the perspective of disaster system theory (Shi et al., 2020), it is also related to vulnerability and human activities. Due to the complexity of damage records themselves, unfortunately, there is no straightforward way to disentangle the fraction that each hazard contributed to the recorded damages, but nevertheless the analysis highlights the importance of joint occurrence with high compound flood potential in causing damages in highly

urbanized areas.

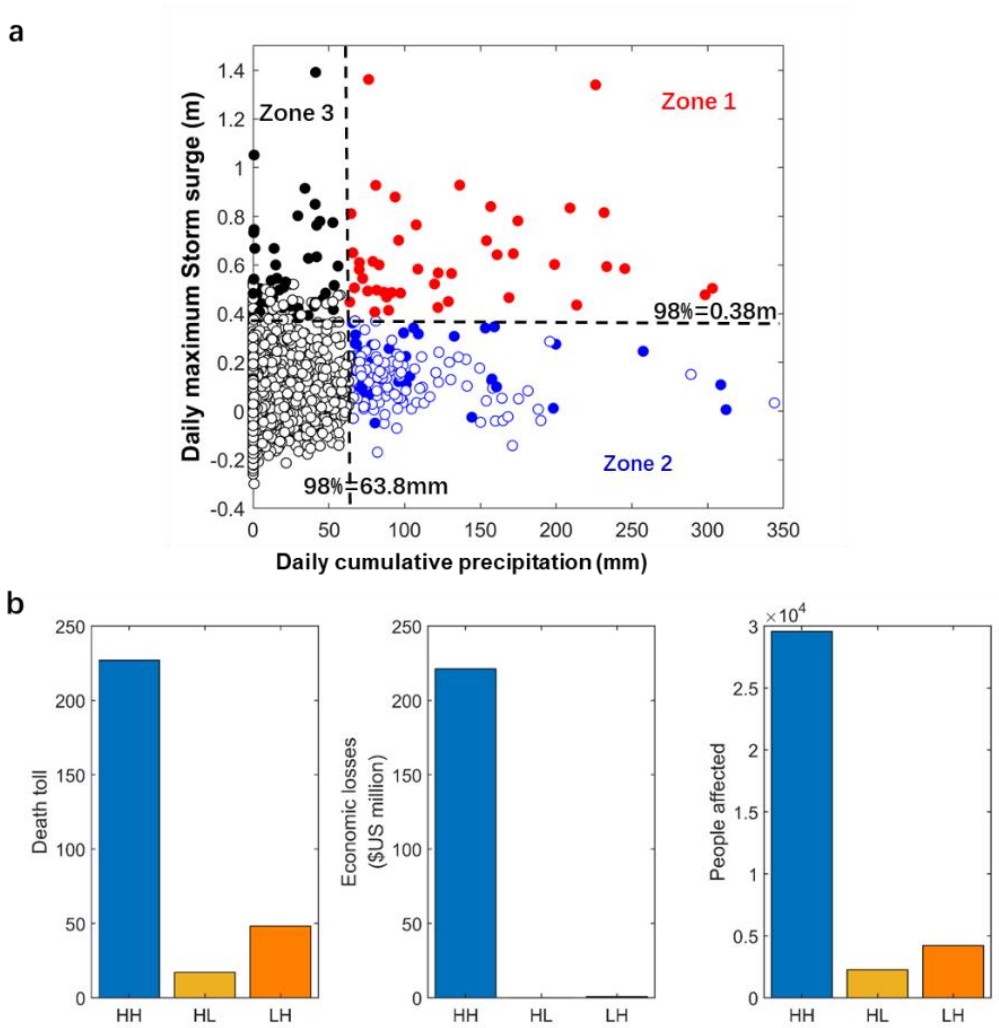

Fig. 9 a) Daily maximum storm surge plotted against daily cumulative precipitation for threshold of 98% for Hong Kong. Filled dots indicate events linked to TC historical damage records. b) damages by joint occurrences (HH: high storm surge and high precipitation) flood and events where only one flood driver was extreme (HL refers to high storm surge and low precipitation; LH refers to low storm surge and high precipitation) in Hong Kong.

## 5 Discussions

The findings presented here are consistent with previous studies conducted for other regions, such as USA (Wahl et al., 2015) and Europe (Ganguli and Merz, 2019). On the one hand, significant dependence exists between various flood hazard drivers which should be taken into consideration when drainage systems and other flood mitigation infrastructure are designed. This is of particular importance for coastal China, as China was the country with the fastest growing amount of artificial impervious areas between 1985-2018 and now ranks first globally in terms of total impervious area (Gong et al., 2019). On the other hand, our results indicate that the frequency of compound events is increasing for coastal China under climate change, in particular sea level rise, which is also in line with previous studies (Moftakhari et al., 2017; Bevacqua et al., 2019). Additional drivers of climate change and variability could further exacerbate the associated flood impacts (Liu et al., 2018). There is evidence, for example, that ENSO has an impact on the dependence between storm surge and precipitation in Australia (Wu and Leonard, 2019). Hence, future research should focus on the interaction between climate processes (e.g., El Nino and/or rising temperatures) and different flooding drivers, such as storm surge, precipitation, river discharge, and waves, and their joint occurrences as well as the associated impacts. The latter are often hard to quantify without using computationally expensive hydrologic and hydraulic models. Eilander et al. (2020) firstly conducted a global analysis forced by a multi-model ensemble of global hydrological models and bounded downstream by a global tide and surge model. Other studies are usually limited to local applications as opposed to larger regional assessments.

One of the main limitations of this study is the relatively small number of tide gauge sites and limited length of the time-series available, especially from TGs. For now, publicly accessible datasets considered here constitute the most comprehensive collection of hourly sea level data along Chinese coasts. There is an urgent need for longer data sets to be used in order to better assess compound flood risk, especially for

south-east China coasts which are prone to TCs. Here we only consider two drivers of flooding, precipitation and storm surge. The role of other flooding drivers needs to be further explored, as well as compound effects under nonstationary conditions, including bivariate frequency analysis, assessing the relationship to climate indices, and the implications for flood risk management. The latter is particularly important, given the low capacity of drainage systems in many Chinese urban areas.

## 6 Conclusions

In this study, we assess the compound flood potential from storm surge and precipitation along major stretches of coastal China. The results show that significant dependence exists between the two flood drivers at many locations, especially at sites in lower latitudes (latitude $< 30°$N). The dependence varies when using different thresholds in the event sampling and is also affected by seasonality. The latter shows that compound events occur more often during the TC season, especially in summer. We also find that sea level rise plays an important role in causing more frequent events with high compound flood potential and it is expected that continuing and accelerating sea level rise will further increase the compound flooding risk. From the perspective of weather patterns, joint occurrence events at south-eastern TG sites (latitude $< 30°$N) are caused by low-pressure systems of similar characteristics carrying intense PWC and causing strong winds that generate storm surges. For Hong Kong, we find that TC events where compound flood potential was high were responsible for the vast majority of the recorded casualties and damages, as opposed to events where compound flooding potential was low because only one driver was extreme.

Ignoring compound effects likely leads to an underestimation of flood risk in coastal China, particularly along the south-eastern coasts. It is therefore crucial that coastal cities and urban planning authorities address compound flood effects (including additional drivers such as river discharge or waves) when designing coastal infrastructure and flood defences or developing adaptation plans to combat the negative impacts of climate change.

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

Acknowledgements. This work is funded by the National Key R & D Program of China (2017YFC1503001); National Natural Science Foundation of China (42001096;42077441); Shanghai Sailing Program (19YF1413700); China Postdoctoral Science Foundation (No. 2019M651429). TW acknowledges funding support from the National Science Foundation (Grant

Number 1929382).

**Author contributions**. FJ (first author) and TW conceived and planned the study. FJ (first author) carried out the analysis and prepared the paper. TW provided guidance on compound flood analysis and contributed to interpretations. FJ (third author), KF, SX and LM offered their expertise in flood, heavy precipitation and hydrometrology, and contributed to revising the

manscript.

**Competing interests**. The authors declare that they have no conflict of interest.

**Data availability**. This study relies entirely on publicly available data from 1) hourly sea level data of 11 TGs with at least

20-year lengths along the Chinese coast from the University of Hawaii Sea Level Center; 2) cumulative daily precipitation records from 1951-2015 are collected from China Meteorological Administration;3) meteorological data from the 20th Century Reanalysis, Version 2c, obtained from the National Oceanic and Atmospheric Administration website; 4) historical damages records from a typhoon database developed by Yap et al. (2015) including historical typhoon records from 1951 to 2012.

**Supplementary.** Sensitivity of threshold on correlation (Supplementary Fig. S1) and Meteorological patterns for events with high compound flood potential and low compound flood potential at the other 9 TGs (not shown in the manuscript) are shown in the supplementary Fig. S2-S10.