# Peer review of "Compound flood potential from storm surge and heavy precipitation in coastal China"

_Hydrology and Earth System Sciences, 2020_

## Referee Comment (RC1) · Anonymous Referee #1 · 31 Jul 2020

Thank you for the opportunity to review this manuscript. This study investigates the compound effects of storm surge and rainfall on coastal floods in China using gauged data from 11 tide gauges. It found that typhoon and sea level rise can potentially increases the frequency of compound coastal floods. In addition, the study attempted to explain the causes of compound events by investigating meteorological forcing. Finally the study concluded that there is a need to incorporate effect of compound floods in risk analysis and infrastructure design. This topic, the method used and the findings are not new. However, it does provide some insights into compound flood risk in China. I have a few comments and suggestions below for the authors to consider.

1.   The authors stated that "To compare impacts caused by compound and non-compound events, we employ a typhoon database developed by Yap et al. (2015),

which includes historical typhoon records from 1951 to 2012, ... ... The database contains information of 853 typhoons in total, with records of direct 115 normalized economic loss (in US$), death toll, and number of people affected". This implies that the authors defined compound coastal flood events as a subset of flood events occurred during typhoon events for impact analysis. Is this categorization correct? Did the authors imply that in China Typhoon is the only cause for compound coastal flood events? Are there any compound flood events occurred outside typhoon events? How the impact of the compound events outside the typhoon events are evaluated or are they included? 2. The damages of compound flood events were assessed using the damages from the typhoon events. However, the damages of typhoon events are not only results of compound flood events embedded in these typhoon events, but also included damages from other effects of these typhoon events. How the impacts of other factors that are not related to compound flood events are isolated or are they included as part of the analysis? 3. It is well known that the threshold selection will have an impact on the dependence analysis, as the authors showed with their results from the sensitivity analysis. Are there any insights derived from this sensitivity analysis that can be used for future analysis, apart from the fact that the results are sensitive to the threshold values used? 4. For seasonal analysis "four periods are considered: typhoon season (July-October), summer (July-August), autumn (September-November), and whole year". Again, this is more related to typhoon events than the defined compound events. 5. Overall, there seems to be a varying definition of "compound flood events" used in the different analysis throughout the paper (e.g. sometimes mixed with typhoon events). This is not only confusing and can be sometimes mis-leading, e.g. for damage analysis commented above. In addition, although various types of analysis were conducted (all of which have been used in previous studies), the manuscript lacks a central theme tying everything together– in other words, why the different types of analysis were selected (apart from the fact that they have been used in similar studies previously) and how they collectively contribute to the understanding of the specific problem under investigation? 6. Finally a minor point: The authors pointed out that

there is a need to assess "the relationship to climate indices". This has been done to some extent. The authors may be interested in the following paper on this topic: Wenyan Wu and Michael Leonard 2019 Impact of ENSO on dependence between extreme rainfall and storm surge Environ. Res. Lett. 14 124043

I hope my comments are helpful for the authors to improve their manuscript.

————————————————————

---

## Referee Comment (RC2) · Anonymous Referee #2 · 17 Aug 2020

This manuscript focuses on compound flood potential from storm surge and heavy precipitation in coastal China, results of which may be a support for urban flood control and management. The idea, the data and the methods used are not new and innovative. The results are common and direct. Two main parts should be improved firstly for further reconsideration for potential publication in HESS. 1. Data are basis for analysis. Tide data collected are mainly from 1975 to 1997 which are not in accord with that of precipitation. Does the tide data in the last 23 years potentially changed under climate change affect the results? If it does, how to improve it? 2. It has been widely accepted that storm surge and heavy precipitation are the first main influence factors of urban flood or waterlogging disasters. Please do not just list the data and their difference, discussions and conclusions must go deeper, mechanism of the results and potential

application in design of flood defences should be clarified.

---

## Referee Comment (RC3) · Anonymous Referee #3 · 26 Aug 2020

This study investigates the compound events from storm surge and heavy precipitation using 11 tide gauges along the coast of China and discusses some potential driving for the occurrences of compound events. This study can provide an important supplement for the analysis of compound events in China owing to the most comprehensive records of storm surge used, even though the methods and results are not very innovative and surprise. There are some concerns that should be addressed for further consideration for potential publication in HESS. Firstly, in the section of "3.1 Selecting compound events", Figure 2 shows the scatter plot for daily maximum storm surge and daily maximum precipitation. You have hourly sea-level data of 11 tide gauge, do you mean to extract the daily maximum one-hour sea level data from these hourly data firstly? But for precipitation data, you only have daily precipitation data, how can you

have daily maximum precipitation? Secondly, in the section "4.2 Effects of sea-level rise on compound event frequencies", it is not very clear how to remove the sea level rise. Do you mean the daily sea level minuses the annual sea level? Thirdly, in the section of "4.5 Impacts caused by compound and non-compound flood events", how can you separate the damages induced by compound events based on typhoon related damages records? For instance, heavy wind due to typhoon events can also result in damages and losses. It is hard to separate the damages from different disasters.

---

## Author Comment (AC1) · 17 Oct 2020

Reviewer 2: This manuscript focuses on compound flood potential from storm surge and heavy precipitation in coastal China, the results of which may be a support for urban flood control and management. The idea, the data and the methods used are not new and innovative. The results are common and direct. Two main parts should be improved firstly for further reconsideration for potential publication in HESS. 1. Data are the basis for analysis. Tide data collected are mainly from 1975 to 1997 which are not in accord with that of precipitation. Does the tide data in the last 23 years potentially changed under climate change affect the results? If it does, how to improve it?

Response: Thanks for the comment. As mentioned in Line 106-108 in the original

manuscript the time series of precipitation observations are usually longer and more complete than tide gauge observations (where data after 1997 is often not publicly available). We are using the longest possible overlapping periods for both datasets in this study. Climate change may have had an impact on compound events over the last two decades. We cannot assess it in detail due to data restrictions, but we make sure to account for the effects of sea level rise in our analysis (by removing its influence through a year-by-year tidal analysis) and further discuss the effects of climate change and variability in the revised version, which includes a new discussion section in response to other reviewer comments (see our comments to them above).

2. It has been widely accepted that storm surge and heavy precipitation are the first main influence factors of urban flood or waterlogging disasters. Please do not just list the data and their difference, discussions and conclusions must go deeper, the mechanism of the results and potential application in design of flood defenses should be clarified.

Response: Thanks for the suggestion. In the revised version, we would like to add a discussion part to have a deeper discussion about impacts by compound events (in parts based on historical damage records) and threats by climate change (particularly sea level rise), and also touch on potential ramifications for the design of flood defenses. However, we also would like to emphasize that no comprehensive analysis of the relationships between the different drivers of compound flooding, and the spatial and temporal variability, has been performed to date for coastal China. Hence, we believe that our results provide a baseline and can guide future research, including more detailed local assessments of the mechanisms through which these compound flooding drivers actually modulate urban flood risk. The latter requires more complex and computationally expensive modelling, which is beyond the scope of our analysis (and likely not possible to perform at the large spatial scales we include in our assessment).

377, 2020.

---

## Author Comment (AC2) · 17 Oct 2020

Reviewer 3:

This study investigates the compound events from storm surge and heavy precipitation using 11 tide gauges along the coast of China and discusses some potential driving for the occurrences of compound events. This study can provide an important supplement for the analysis of compound events in China owing to the most comprehensive records of storm surge used, even though the methods and results are not very innovative and surprise. There are some concerns that should be addressed for further consideration for potential publication in HESS. Firstly, in the section of "3.1 Selecting compound events", Figure 2 shows the scatter plot for daily maximum storm surge and

daily maximum precipitation. You have hourly sea-level data of 11 tide gauge, do you mean to extract the daily maximum one-hour sea level data from these hourly data firstly? But for precipitation data, you only have daily precipitation data, how can you have daily maximum precipitation?

Response: Thanks for the comment. We are sorry for the confusion. Firstly, we apply a harmonic tidal analysis by using hourly sea level observations to extract the surge (or non-tidal residual) part. Then, we extract the daily maximum surge from hourly surge data. For daily precipitation data, it is the amount of accumulated daily precipitation. This is clarified in the revised version.

Secondly, in the section "4.2 Effects of sea-level rise on compound event frequencies", it is not very clear how to remove the sea level rise. Do you mean the daily sea level minuses the annual sea level?

Response: We removed the mean sea level influence by applying a year-by-year harmonic tidal analysis (see Line 100). In doing so we effectively remove the tidal influence but also the annual mean sea level from the hourly (and daily maxima) storm surge data which is ultimately used in the analysis. This is the same approach used in many previous studies and we will make it clearer in the revised version of the paper.

Thirdly, in the section of "4.5 Impacts caused by compound and non-compound flood events", how can you separate the damages induced by compound events based on typhoon related damages records? For instance, heavy wind due to typhoon events can also result in damages and losses. It is hard to separate the damages from different disasters.

Response: Thanks for the comment. The reviewer raised a very good point. The damages records developed by Yap et al. (2015) is the total damages by more than one hazard. It may be caused by one hazard, or two or more hazards, such as gale, heavy rainfall and storm surge. From the perspective of disaster system theory, it is also related to vulnerability and human activities. Unfortunately, there is no straightforward

way to disentangle the fraction that each hazard contributed to the damage. In this case, we would like to show the difference caused by compound and non-compound events, assuming that flooding was the main contributor to the damages or at least had a similar relative contribution to the damages. We realize that this is big assumption to make and based on the reviewer's comment (and similar comments from another reviewer) we decided to move this part into a new discussion section where the underlying issues are discussed when attempting to link compound and non-compound events to the damage database.

---

## Author Response (AR1)

Dear reviewers and editor,

First, we would like to thank you for the possibility to revise the paper HESS-2020-377 and for your consideration of its publication in Hydrology and Earth System Sciences. We greatly appreciate the careful reading of the paper and the constructive and precise comments that significantly helped to improve major portions of the manuscript. Main comments are raised from the historical damage analysis part, thus, we did major changes for it, especially in the Discussion part. We were keen to integrate all comments highlighted by the reviewers, evidenced by our comments on the reviewers' main points below. We maintained the order of comments as provided by the editor to facilitate reviewers finding their own comments.
* * *
Reviewer 1:

*Thank you for the opportunity to review this manuscript. This study investigates the compound effects of storm surge and rainfall on coastal floods in China using gauged data from 11 tide gauges. It found that typhoon and sea level rise can potentially increases the frequency of compound coastal floods. In addition, the study attempted to explain the causes of compound events by investigating meteorological forcing. Finally the study concluded that there is a need to incorporate effect of compound floods in risk analysis and infrastructure design. This topic, the method used and the findings are not new. However, it does provide some insights into compound flood risk in China.*

**Response**: Thanks for the comment. This is indeed the first time such a comprehensive study of compound flooding is carried out for coastal China; in addition to findings highlighted by the reviewer we also feel the results regarding the variation across seasons and regions are interesting and relevant.

*I have a few comments and suggestions below for the authors to consider. 1. The authors stated that "To compare impacts caused by compound and noncompound events, we employ a typhoon database developed by Yap et al. (2015), which includes historical typhoon records from 1951 to 2012, ......The database contains information of 853 typhoons in total, with records of direct 115 normalized economic loss (in US$), death toll, and number of people affected". This implies that the authors defined compound coastal flood events as a subset of flood events occurred during typhoon events for impact analysis. Is this categorization correct? Did the authors imply that in China Typhoon is the only cause for compound coastal flood events? Are there any compound flood events occurred outside typhoon events? How the impact of the compound events outside the typhoon events are evaluated or are they included?*

**Response:** Thanks for the comment. We agree with the reviewer that compound events outside the Typhoon season may be excluded, but also stress that in China, typhoon is the main cause for compound coastal flood events. As shown in the replotted Fig. 8a, all compound events (high storm surge and high precipitation at 98% thresholds) for Hong Kong were linked to typhoon records. If we assume 97% is the threshold, there were some compound events outside typhoon events. Not all those events lead to significant damages or the damages were not recorded. In this case, we would like to show the difference caused by compound and non-compound events, assuming that flooding was the main contributor to the damages or at least had a similar relative contribution to the damages.

Historical damage records are sparse and often unavailable to us; in addition it is of course tricky to match the damages to compound flood events. The reasons can be summarized into the following categories: 1) limited time series: most datasets we found have records after 1985. However, the observations are mainly between 1975 and 1997. It makes the damage record incompatible with observations. 2) incomplete information: most of them are annual flood damage records, with only one record for each year without information of occurrence time. It is not enough information to match the damages with compound events we extract from observations. 3) poor quality: global flood damage datasets (EMDAT for example) are too coarse to be useful for our purposes. Other datasets are raw damage reports without quality control. Thus, this dataset developed by Yap et al. (2015) is the most feasible for us to use at the moment.

The reviewer's comment has prompted us to reframe the way the damage dataset is considered, which includes changing some of the statements/conclusions we draw from it, and also moving it from the results section into a new discussion section, where we touch on some specific cases and the shortcomings outlined here (and in the next comment) are also addressed; we use this to highlight the necessity for better damage information to be made available and stress that our analysis is only a first step and could be used as a baseline in future research.
→*See Section 5 Discussion Line 305-335 in the clean version.*

*2. The damages of compound flood events were assessed using the damages from the typhoon events. However, the damages of typhoon events are not only results of compound flood events embedded in these typhoon events, but also included damages from other effects of these typhoon events. How the impacts of other factors that are not related to compound flood events are isolated or are they included as part of the analysis?*

**Response:** The reviewer raised a very good point, which is also pointed out by the third reviewer. We agree with the reviewer that typhoon damage records may also include effects like gale, storm surge, precipitation, etc. Unfortunately, there is no straightforward way to disentangle the fraction that each hazard contributed to the damage. Thus, we decided to still include parts of the damage analysis, but instead of showing it in the results we moved it to the discussion and explain the underlying uncertainties/issues.
→*See Section 5 Discussion.*

*3. It is well known that the threshold selection will have an impact on the dependence analysis, as the authors showed with their results from the sensitivity analysis. Are there any insights derived from this sensitivity analysis that can be used for future analysis, apart from the fact that the results are sensitive to the threshold values used?*

**Response:** Thanks for the suggestion. We tried to draw such conclusions in the beginning, but realized it is difficult to draw generalizable insights from this sensitivity analysis. It is very localized and highly dependent on the underlying data. There are other methods to represent bivariate extremes (e.g. Salvadori, et al. (2016), A multivariate copula-based framework for dealing with

hazard scenarios and failure probabilities. Water Resources Research, 52(5), pp.3701-3721.), which we didn't employ here as it would go beyond the scope of our study.
→See Section 4.1 Line 183-186 in the clean version.

*4. For seasonal analysis "four periods are considered: typhoon season (July-October), summer (July-August), autumn (September-November), and whole year". Again, this is more related to typhoon events than the defined compound events.*

**Response:** In this study, we select these three seasons to show seasonal variation. Our hypothesis is that there will be seasonal variation in compound flood frequency, with some coastal regions experiencing a greater dependency in summer or in typhoon seasons. We first sample all compound events, then select compound events which happened in these three seasons, to calculate their dependence. It helps to understand in which season the likelihood of compound events to occur is relatively higher.

*5. Overall, there seems to be a varying definition of "compound flood events" used in the different analysis throughout the paper (e.g. sometimes mixed with typhoon events). This is not only confusing and can be sometimes mis-leading, e.g. for damage analysis commented above.*

**Response:** Sorry for the confusion, in the revised version make it clearer why events are selected the way they are for the different analysis steps. Typhoons are the leading cause for compound flooding events and hence we pay particular attention to these. The point regarding the damage analysis is addressed in our responses above (and also in our comments for reviewer #3).

*In addition, although various types of analysis were conducted (all of which have been used in previous studies), the manuscript lacks a central theme tying everything together– in other words, why the different types of analysis were selected (apart from the fact that they have been used in similar studies previously) and how they collectively contribute to the understanding of the specific problem under investigation?*

**Response:** Thanks for the suggestion.
The reason for selecting different types of compound events depends on the purpose of analysis as well as sample sizes. The core definition could be seen in Fig. 2 in the manuscript. The reason for selecting Case1 and Case2 could been seen in Wahl et al. (2015) and also in Line 120-130. Case1 and Case2 could be separated into three zones (see Hendry et al. 2019 and Zheng et al., 2013).
In the revised version, we changed the title to "Assessing the characteristics and drivers of compound flood events from storm surge and precipitation in coastal China". The manuscript has three objectives: 1) identify and collate compound events from storm surge and precipitation, and analyse their dependence; 2) examine how the strength of dependence between storm surge and precipitation are influenced by seasons and threshold selection; 3) identify the driving weather patterns of compound/non-compound events. We believe that addressing these 3 objectives in concert reflects our overarching goal related to the (new) title, and the conclusion section will be reworked accordingly. As outlined above, we add a new discussion section where we make the transition from focusing on the dependence, it's variability across seasons and regions, and the

driving weather patterns to the impacts caused by compound events (based on historical damage records).

*→See Section 1 Line 88-91 in the clean version.*

Hendry, A., Haigh, I., Nicholls, R., Winter, H. and Neal, R., 2019, April. Assessing the characteristics and likelihood of compound flooding events around the UK. Hydrol. Earth Syst. Sci., 23, 3117–3139.

Wahl, T., Jain, S., Bender, J., Meyers, S.D. and Luther, M.E., 2015. Increasing risk of compound flooding from storm surge and rainfall for major US cities. Nature Climate Change, 5(12), p.1093.

Zheng, F., Westra, S., Sisson, & S., A. 2013. Quantifying the dependence between extreme rainfall and storm surge in the coastal zone. Journal of Hydrology. 505. pp.172-187

*6. Finally a minor point: The authors pointed out that there is a need to assess "the relationship to climate indices". This has been done to some extent. The authors may be interested in the following paper on this topic: Wenyan Wu and Michael Leonard 2019 Impact of ENSO on dependence between extreme rainfall and storm surge Environ. Res. Lett. 14 124043.I hope my comments are helpful for the authors to improve their manuscript.*

**Response:** Thanks for sharing. This is included now in the discussion, and it would be interesting to carry out a similar analysis in the future. We have added related references in the discussion part. *→See Section 5 Discussion Line295-300 in the clean version.*

Reviewer 2:

*This manuscript focuses on compound flood potential from storm surge and heavy precipitation in coastal China, results of which may be a support for urban flood control and management. The idea, the data and the methods used are not new and innovative. The results are common and direct. Two main parts should be improved firstly for further reconsideration for potential publication in HESS.*
*1. Data are basis for analysis. Tide data collected are mainly from 1975 to 1997 which are not in accord with that of precipitation. Does the tide data in the last 23 years potentially changed under climate change affect the results? If it does, how to improve it?*

**Response**: Thanks for the comment.

As mentioned in Line 106-108 in the original manuscript the time series of precipitation observations are usually longer and more complete than tide gauge observations (where data after 1997 is often not publicly available). We are using the longest possible overlapping periods for both datasets in this study.

Climate change may have an impact on the tides (through sea level rise), but over the timescales analyzed here it would be negligible (see for example recent review paper by Haigh et al., "The tides they are a-changing", https://doi.org/10.1029/2018RG000636) . We have considered the effect of sea level rise explicitly in our analysis (by removing it's influence through a year-by-year tidal analysis). We also found there is an increase of compound events when mean sea level rise is included (see Section 4.2).

*2. It has been widely accepted that storm surge and heavy precipitation are the first main influence factors of urban flood or waterlogging disasters. Please do not just list the data and their difference, discussions and conclusions must go deeper, mechanism of the results and potential application in design of flood defenses should be clarified.*

**Response**: Thanks for the suggestion. In the revised version, we added a discussion part to have a deeper discussion about impacts by compound events (in parts based on historical damage records) and threats by climate change drivers, and also touch on potential ramifications for the design of flood defenses.

→*See Section 5 Discussion.*

Reviewer 3:

*This study investigates the compound events from storm surge and heavy precipitation using 11 tide gauges along the coast of China and discusses some potential driving for the occurrences of compound events. This study can provide an important supplement for the analysis of compound events in China owing to the most comprehensive records of storm surge used, even though the methods and results are not very innovative and surprise. There are some concerns that should be addressed for further consideration for potential publication in HESS. Firstly, in the section of "3.1 Selecting compound events", Figure 2 shows the scatter plot for daily maximum storm surge and daily maximum precipitation. You have hourly sea-level data of 11 tide gauge, do you mean to extract the daily maximum one-hour sea level data from these hourly data firstly? But for precipitation data, you only have daily precipitation data, how can you have daily maximum precipitation?*

**Response**: Thanks for the comment. We are sorry for the confusion. Firstly, we apply a harmonic tidal analysis by using hourly sea level observations to extract the surge (or non-tidal residual) part. Then, we extract the daily maximum surge from hourly surge data. For daily precipitation data, it is the amount of accumulated daily precipitation. This is clarified in the revised version.

→*See Section 2.*

*Secondly, in the section "4.2 Effects of sea-level rise on compound event frequencies", it is not very clear how to remove the sea level rise. Do you mean the daily sea level minuses the annual sea level?*

**Response**: We removed the mean sea level influence by applying a year-by-year harmonic tidal analysis (see Line 100). In doing so we effectively remove the tidal influence but also the annual mean sea level from the hourly (and daily maxima) storm surge data which is ultimately used in the analysis. This is the same approach used in many previous studies and we will make it clearer in the revised version of the paper.

→*See Section 2.*

*Thirdly, in the section of "4.5 Impacts caused by compound and non-compound flood events", how can you separate the damages induced by compound events based on typhoon related damages records? For instance, heavy wind due to typhoon events can also result in damages and losses. It is hard to separate the damages from different disasters.*

**Response**: Thanks for the comment.

We agree with the reviewer that there is no straightforward way to disentangle the fraction that each hazard contributed to the damage. In this case, we would like to show the difference caused by compound and non-compound events, assuming that flooding was the main contributor to the damages or at least had a similar relative contribution to the damages. We realize that this is a big assumption to make and based on the reviewer's comment (and similar comments from the first reviewers) we moved this part into a new discussion section where the underlying issues are discussed when attempting to link compound and non-compound events to the damage database.

→*See Section 5 Discussion.*

---

## Referee Report (RR1)

The authors present a detailed analysis of various indicators typically used to assess the compound flood potential for 11 locations along the coast of China. It is clear after reading the manuscript that the authors have put a substantial amount of work in the execution of the methodology: selecting different thresholds to quantify the statistical dependence, looking at the influence of seasonality, sea-level rise and weather patterns for marginal or joint extremes of storm surge and precipitation. The main objectives of this paper do not appear scientifically novel to me but rather a thorough application of current methods. If the goal of the paper is to provide new insights on the compound flood potential in China, I would discuss this further in the discussion and highlight in the conclusion how your findings complement or contrast results from other local or global studies. Alternatively, another journal like NHESS could be more suitable to report such findings since I think that the fact that the paper focuses on the Asian coastline is a particularly relevant point for risk assessments. I listed below a few major and minor comments for the authors to consider.

Major comments:
- Throughout the paper, it seems that the authors interchangeably use the terms 'flood' and 'event' which is very confusing. In Figure 2, the authors clearly state what they define as a compound or non-compound event. However, these events (points on Figure 2) do not necessarily generate floods. Yet, this confusion is omnipresent throughout the manuscript, for example:
    - The title mentions "compound flood events" whereas the abstract mentions that "This paper investigates the potential compound effects". Those statements have very different implications when interpreting the results and conclusion.
    - The authors mention three different definitions of 'compound events': the first one from Zscheischler et al. (2018), the second one from Wahl et al (2015) and the third one suggested by the authors in Figure 2. These three definitions are different so the authors should be clear about this and discuss the limitations of this selection in the discussion section. As correctly mentioned in the introduction, Zscheischler et al. (2018) could consider any combinations (also both non-extreme) to be a compound flood: selection has to be done based on impact (which is not known in this case). Wahl et al (2015) would consider any points in Zone 1, 2 or 3 to be a compound flood. I appreciate the fact that the authors are clear in the text and always mentioning when they refer to Zones 1/2/3 vs Case 1/2 but it becomes very confusing when interpreting results and conclusions in terms of compound and non-compound. The dependence and frequency analysis is done in terms of Case 1/2 but the weather maps and analysis of the typhoon dataset are done in terms of the Zones. In both sections, the terms 'compound' is used but I am not sure anymore what it really means as it refers to different areas in Figure 2. Clearly there is some value in this analysis but the discussion and conclusion have to be carefully rephrased to express the limitations of these definitions.
- The analysis based on the typhoon database is interesting but highly uncertain, especially when generalized with respect to compound/non-compound events. The authors

acknowledge that convective rainfall events are probably excluded from the typhoon dataset but no information on the damage from these events caused is added. Yet, conclusions about compound/non-compound flood events are made. As the authors state on line 316, we do not know whether those events lead to no damage or significant damage. This could lead to very different conclusions than the ones presented here. I find this analysis interesting but I would recommend acknowledging the fact that you focus only on typhoons for this analysis and instead show the influence of both drivers on damages when only considering typhoons, and not generalize it to compound/non-compound events.

- Did the authors consider comparing their results based on skew surge instead of storm surge? When performing a tidal analysis, small errors in the phase of the tide can lead to large storm surge peaks. This could have a large influence on your correlation. The authors mention on line 106 that the data has been checked for common errors but do not elaborate further.

Other comments:

- Convective rainfall is discussed is the discussion section (section 5) but is actually not mentioned when describing the weather patterns (section 4.4). I would recommend introducing this weather pattern earlier if you mention it for Hong Kong, this will help the reader understand all weather systems conducive to flooding.
- The authors mention on lines 303-305 that few regional assessments from hydrodynamic models have been conducted for compound flooding. Such analysis has been conducted at the global scale and it could be interesting to comment on this with respect to the patterns found in your study:

  *Eilander, D., Couasnon, A., Ikeuchi, H., Muis, S., Yamazaki, D., Winsemius, H. C., & Ward, P. J. (2020). The effect of surge on riverine flood hazard and impact in deltas globally. Environmental Research Letters, 15(10), 104007.*
- Some limitations are discussed in the conclusions (paragraph starting in line 350). I would move those points and elaborate them in a separate section or combine it with the discussion. Similarly, I would say that the analysis of the typhoon database in the discussion belongs more to the result section than discussion.
- Line 96: "where tropical cyclones impacts are more severe". I am not sure why it is important to mention this here. Maybe make this clearer and/or add reference to support this because this is not clear to me when looking at Figure 1.
- Line 107: what do the authors mean by "earlier" here?
- Line 115: This is minor but it would be more logical to write sea *level* pressure for SLP instead of sea *surface* pressure
- Line 118: Maybe use the term "Defining" instead of "Selecting" as compound events are described in various ways in this paper.
- Line 137: Maybe change the word "appropriate". Both annual maxima and POT can be used in this type of analysis as shown by previous literature.
- On Figure 4, I suggest changing the label of the colorbar to highlight that it is a difference. Otherwise, the negative values seem strange at first sight.

- Line 210: "To better understand the timing of events leading to joint dependence throughout the year". This sentence is not clear to me. I would suggest rephrasing it.
- Line 239-240: "The summer monsoon brings continuous precipitation since June to August in southern China. Thus, the dependence is higher in the summer compared to the typhoon season". Does this conclusion applies to all the gauges or only the last ones discussed (TG7 and TG10)? It would be useful to elaborate a bit more because I am not sure I understand this as currently phrased. Does the summer monsoon also generate storm surge? If the dependence is higher, this implies that storm surge and precipitation are more strongly correlated. If only the rainfall is higher but the storm surge is random, the correlation will be insignificant.
- Line 326: explain "gale" briefly?
- I would suggest labeling the gauges again on Figure 3b. This makes comparison of both panels a and b easier.
- I would strongly recommend carefully checking the manuscript for typos and other mistakes. Below are a few examples I found:
  - Line 89, 131, 132, 232: spaces missing
  - The description of the zones is sometimes flipped with what is shown on Figure 2. For example in the description of Figure 2, I think it should be "i.e. high precipitation and high storm surge, respectively"(line 134). This is also the case on line 162.
  - In Figure 2 and 8, the x-axis label should be "Precipitation"
  - Line 319: I think the word average is missing in (US$ 5 million per event)?
  - Line 334: remove 'were' or add 'that'

---

## Author Response (AR2)

Dear reviewers and editor,

Thanks for the possibility to revise the paper HESS-2020-377 and to the reviewers for providing very useful comments which helped us to improve the manuscript. We have updated most of the figures and rearranged the manuscript structure according to the reviewer feedback, also format of references. Our responses to each reviewer comment in turn are listed below (in blue text).
* * *
**Editor comment:**

*your manuscript has now been refereed by one of the original reviewers and one additional reviewer. You will see that some open points remain that require further revisions before I can make a final decision. Furthermore, one of the reviewers criticised that the title is very similar to that of Hendry et al. (2019) already published in HESS and mentioned that this maybe done on purpose but it also indicates the lack of scientific novelty in this contribution. Therefore, in your revised version please consider changing the title to highlight which novel aspects your contribution contains.*

**Response**: Thanks. We have changed title to "Compound flood potential from storm surge and heavy precipitation in coastal China: dependence, drivers, and impacts". We updated most of the figures, rearranged the manuscript structure (four methods and four result section) and rephrased the related statements to match the new title.

**Reviewer 1:**

*The authors present a detailed analysis of various indicators typically used to assess the compound flood potential for 11 locations along the coast of China. It is clear after reading the manuscript that the authors have put a substantial amount of work in the execution of the methodology: selecting different thresholds to quantify the statistical dependence, looking at the influence of seasonality, sea-level rise and weather patterns for marginal or joint extremes of storm surge and precipitation. The main objectives of this paper do not appear scientifically novel to me but rather a thorough application of current methods. If the goal of the paper is to provide new insights on the compound flood potential in China, I would discuss this further in the discussion and highlight in the conclusion how your findings complement or contrast results from other local or global studies. Alternatively, another journal like NHESS could be more suitable to report such findings since I think that the fact that the paper focuses on the Asian coastline is a particularly relevant point for risk assessments. I listed below a few major and minor comments for the authors to consider.*

**Response**: Thanks for the comment. In this revised version, we updated most of the figures and rearranged the manuscript structure. Here we focus on three aspects of compound flood potential from storm surge and heavy precipitation in coastal China:(1) dependence between driver combinations, seasonal variations and threshold selection, (2) role of sea level rise and meteorology patterns, and (3) possible impacts of compound events. It also matches the new title. We are now comparing our findings with other local or global studies in the results section (such as Line 261-270) and the discussion section (Line 346-350).

*Major comments:*

*Throughout the paper, it seems that the authors interchangeably use the terms 'flood' and 'event' which is very confusing. In Figure 2, the authors clearly state what they define as a compound or non-compound event. However, these events (points on Figure 2) do not necessarily generate floods.*

*Yet, this confusion is omnipresent throughout the manuscript, for example:*

- *The title mentions "compound flood events" whereas the abstract mentions that "This paper investigates the potential compound effects". Those statements have very different implications when interpreting the results and conclusion.*
- *The authors mention three different definitions of 'compound events': the first one from Zscheischler et al. (2018), the second one from Wahl et al (2015) and the third one suggested by the authors in Figure 2. These three definitions are different so the authors should be clear about this and discuss the limitations of this selection in the discussion section. As correctly mentioned in the introduction, Zscheischler et al. (2018) could consider any combinations (also both non-extreme) to be a compound flood: selection has to be done based on impact (which is not known in this case). Wahl et al (2015) would consider any points in Zone 1, 2 or 3 to be a compound flood. I appreciate the fact that the authors are clear in the text and always mentioning when they refer to Zones 1/2/3 vs Case 1/2 but it becomes very confusing when interpreting results and conclusions in terms of compound and non-compound. The dependence and frequency analysis is done in terms of Case 1/2 but the weather maps and analysis of the typhoon dataset are done in terms of the Zones. In both sections, the terms 'compound' is used but I am not sure anymore what it really means as it refers to different areas in Figure 2. Clearly there is some value in this analysis but the discussion and conclusion have to be carefully rephrased to express the limitations of these definitions.*

**Response:** Thanks for the comment. The reviewer raised a very good point which needs to be carefully noticed. We acknowledge that there are different ways to define compound events. In this revised version, we rephrased the statement clearly. We assume compound events here are combination of storm surge and precipitation with at least one extreme variable. We also changed how Case1 and Case 2 are considered; we now focus on Zones 1, 2 and 3 which is also consistent with the weather type analysis and the damage assessment. We checked the use the term "compound" throughout the manuscript to make sure it is consistent.

→*See Section 3.1 Line 129-132 and Line 139-146.*

*The analysis based on the typhoon database is interesting but highly uncertain, especially when generalized with respect to compound/non-compound events. The authors acknowledge that convective rainfall events are probably excluded from the typhoon dataset but no information on the damage from these events caused is added. Yet, conclusions about compound/non-compound flood events are made. As the authors state on line 316, we do not know whether those events lead to no damage or significant damage. This could lead to very different conclusions than the ones presented here. I find this analysis interesting but I would recommend acknowledging the fact that you focus only on typhoons for this analysis and instead show the influence of both drivers on damages when only considering typhoons, and not generalize it to compound/non-compound events.*

**Response:** Thanks. Following the previous, we have changed the compound/non-compound statement throughout the manuscript. Here we try to show the differences of impacts caused by the joint occurrence (two extremes happen at the same time) and one extreme event. We acknowledge that high uncertainty exists in the manuscript

→*See Section 4.4* (Line 331-340).

*Did the authors consider comparing their results based on skew surge instead of storm surge? When performing a tidal analysis, small errors in the phase of the tide can lead to large storm surge peaks. This could have a large influence on your correlation. The authors mention on line 106 that the data has been checked for common errors but do not elaborate further.*

**Response:** Thanks for the comment. We are aware that several previous studies used skew surge instead of storm surge. We have not employed it here but assume that while it might have an effect for individual surge events (if the errors outlined by the reviewer indeed exist in the data we used), but we would expect the influence on the overall conclusions to be negligible.

Other comments:

• *Convective rainfall is discussed is the discussion section (section 5) but is actually not mentioned when describing the weather patterns (section 4.4). I would recommend introducing this weather pattern earlier if you mention it for Hong Kong, this will help the reader understand all weather systems conducive to flooding.*

**Response:** Thanks for the comment. We now introduce it in the section focusing on weather patterns**.**

• *The authors mention on lines 303-305 that few regional assessments from hydrodynamic models have been conducted for compound flooding. Such analysis has been conducted at the global scale and it could be interesting to comment on this with respect to the patterns found in your study: Eilander, D., Couasnon, A., Ikeuchi, H., Muis, S., Yamazaki, D., Winsemius, H. C., & Ward, P. J. (2020). The effect of surge on riverine flood hazard and impact in deltas globally. Environmental Research Letters, 15(10), 104007.*

**Response:** Thanks for sharing this work; it was published after our first submission, thus we neglected it before, but now include it in the discussion section.

• *Some limitations are discussed in the conclusions (paragraph starting in line 350). I would move those points and elaborate them in a separate section or combine it with the discussion. Similarly, I would say that the analysis of the typhoon database in the discussion belongs more to the result section than discussion.*

**Response:** We moved the analysis of the typhoon database to results section (section 4.4) and revised the discussion.

• *Line 96: "where tropical cyclones impacts are more severe". I am not sure why it is important to mention this here. Maybe make this clearer and/or add reference to support this because this is not clear to me when looking at Figure 1.*

**Response:** Thanks, we added the reference to support it.

• *Line 107: what do the authors mean by "earlier" here?*

**Response:** For Hong Kong, sea level was recorded at North Point between 1962 and 1986 and then moved to Quarrybay. The offset between the two records is 1.02 cm. We combined these two data after shifting the earlier data by 1.02 cm. It has been described in Ding et al. (2002).

• *Line 115: This is minor but it would be more logical to write sea* level *pressure for SLP instead of sea* surface *pressure*

● *Line 118: Maybe use the term "Defining" instead of "Selecting" as compound events are described in various ways in this paper.*

●*Line 137: Maybe change the word "appropriate". Both annual maxima and POT can be used in this type of analysis as shown by previous literature.*

**Response:** Thanks, we changed above three comments accordingly.

● *On Figure 4, I suggest changing the label of the colorbar to highlight that it is a difference. Otherwise, the negative values seem strange at first sight.*

**Response:** Thanks, we have replotted figure 4.

●*Line 210: "To better understand the timing of events leading to joint dependence throughout the year". This sentence is not clear to me. I would suggest rephrasing it.*

**Response:** We have rephrased it.

● *Line 239-240: "The summer monsoon brings continuous precipitation since June to August in southern China. Thus, the dependence is higher in the summer compared to the typhoon season". Does this conclusion applies to all the gauges or only the last ones discussed (TG7 and TG10)? It would be useful to elaborate a bit more because I am not sure I understand this as currently phrased. Does the summer monsoon also generate storm surge? If the dependence is higher, this implies that storm surge and precipitation are more strongly correlated. If only the rainfall is higher but the storm surge is random, the correlation will be insignificant.*

**Response:** The conclusion applies to the south-east TGs. Two months in the summer season, namely July and August, overlap with typhoon season. During typhoon season, storm surge is more frequent. With continuous precipitation brought by the summer monsoon, it is likely to generate compounding effects.

●*Line 326: explain "gale" briefly?*

**Response:** Gale here refers to strong wind caused by Typhoon, it also could lead to damage in the duration of typhoon event. This information has been added.

●*I would suggest labeling the gauges again on Figure 3b. This makes comparison of both panels a and b easier.*

**Response:** Thanks, we changed it and the following figures accordingly.

●*I would strongly recommend carefully checking the manuscript for typos and other mistakes. Below are a few examples I found:*

   *o Line 89, 131, 132, 232: spaces missing*

   *o The description of the zones is sometimes flipped with what is shown on Figure 2. For example in the description of Figure 2, I think it should be "i.e. high precipitation and high storm surge, respectively"(line 134). This is also the case on line 162.*

   *o Line 319: I think the word average is missing in (US$ 5 million per event)?*

   *o Line 334: remove 'were' or add 'that'*

**Response:** Thanks, we changed above comments accordingly.

*o In Figure 2 and 8, the x-axis label should be "Precipitation"*

**Response:** We now use "Daily Cumulative Precipitation" to make it clear.

**Reviewer 2:**

*The manuscript titled "Assessing the characteristics and drivers of compound flood events from storm surge and heavy precipitation in coastal China" aims to identify compound events from storm surge and precipitation, analyses their dependence during different seasons and using threshold selection, and examine the potential weather patterns conducive to compound events. The revised manuscript is well written and has addressed my concerns. I only have a comment about the harmonic tidal analysis method before it is accepted for publication.*

*Line 103-Line 195, authors apply a harmonic tidal analysis to remove influences of mean sea level. However, most common readers are not familiar with the method. I would recommend authors add more details about the process and provide a schematic diagram to show the original time series and the time series after removing the long-trend trend, year-to-year, decadal variability, respectively.*

**Response:** Thanks for the comment. This method is really standard and has been used in so many previous studies that we feel adding a detailed explanation and even a figure would distract from the main focus of our analysis.

*Paragraph 89, typo "analysetheir"*

**Response:** Thanks for pointing out, changed accordingly.